# Investigation of Calcium and Magnesium Removal by Donnan Dialysis According to the Doehlert Design for Softening Different Water Types

**DOI:** 10.3390/membranes13020203

**Published:** 2023-02-07

**Authors:** Ikhlass Marzouk-Trifi, Lassaad Baklouti, Lasâad Dammak

**Affiliations:** 1Laboratoire de Recherche Dessalement ET Traitement Des Eaux, Faculté Des Sciences de Tunis, Université de Tunis El Manar, Tunis 1068, Tunisia; 2Department of Chemistry, College of Sciences and Arts at ArRass, Qassim University, Arras 51921, Saudi Arabia; 3Université Paris-Est Créteil, CNRS, ICMPE, UMR 7182, 2 rue Henri Dunant, 94320 Thiais, France

**Keywords:** hardness, Donnan Dialysis, water softening, response surface methodology

## Abstract

In this study, calcium and magnesium were removed from Tunisian dam, lake, and tap water using Donnan Dialysis (DD) according to the Doehlert design. Three cation-exchange membranes (CMV, CMX, and CMS) were used in a preliminary investigation to establish the upper and lower bounds of each parameter and to more precisely pinpoint the optimal value. The concentration of compensating sodium ions [Na^+^] in the receiver compartment, the concentration of calcium [Ca^2+^] and magnesium [Mg^2+^] in the feed compartment, and the membrane nature were the experimental parameters. The findings indicate that the CMV membrane offers the highest elimination rate of calcium and magnesium. The Full Factorial Design makes it possible to determine how the experimental factors affect the removal of calcium and magnesium by DD. All parameters used had a favorable impact on the response; however, the calcium and magnesium concentration were the most significant ones. The Doehlert design’s Response Surface Methodology (RSM) was used to determine the optimum conditions ([Mg^2+^] = 90 mg·L^−1^, [Ca^2+^] = 88 mg·L^−1^, [Na^+^] = 0.68 mol·L^−1^) allowing a 90.6% hardness removal rate with the CMV membrane. Finally, we used Donnan Dialysis to remove calcium and magnesium from the three different types of natural water: Dam, Lake, and Tap water. The results indicate that, when compared to lake water and tap water, the removal of calcium and magnesium from dam water is the best. This can be linked to the water matrix’s complexity. Therefore, using Donnan Dialysis to decrease natural waters hardness was revealed to be suitable.

## 1. Introduction

Historically, Tunisia is considered an arid and semiarid region due to its limited freshwater resources, making brackish, drainage, and surface water essential sources for drinking. There are issues related to the quantity and quality of Tunisia’s surface water resources. Due to the semi-arid to the arid climate found in most parts of the country, periodic droughts, and salty rocks found within the country, water quality has naturally deteriorated [1], making these water resources not immediately reusable because of hardness [2]. Calcium and magnesium ions are mainly responsible for water hardness, and to a lesser extent, the presence of iron and manganese. Water hardness is formed when water percolates through deposits of limestone, chalk, or gypsum [3]. Water is usually classified as soft (0–60 mg·L^−1^), moderately hard (60–120 mg·L^−1^), hard (120–180 mg·L^−1^), and very hard (>180 mg·L^−1^) [4]. Water hardness higher than 200 mg·L^−1^ can be tolerated but when it is higher than 500 mg·L^−1^, it is no longer suitable for consumption. Water hardness may have moderate health benefits [5]. Nerbrand et al. [6] evaluated the relation between calcium and magnesium in drinking water and risk factors for cardiovascular disease in individuals living in hard and soft water areas with considerable differences in cardiovascular mortality. Results revealed significant correlations between the amount of calcium and magnesium in water and major cardiovascular risk factors. Hardness also poses critical problems in industrial settings, where water hardness is monitored to avoid costly breakdowns in boilers, cooling towers, and other equipment that handles water [7]. Many techniques have been used for the removal of calcium and magnesium from water, wastewater, and seawater, such as chemical precipitation [8,9], electrochemical precipitation [10,11], adsorption [12,13], electrolysis [14,15], electrodialysis [16,17], reverse osmosis [18], nanofiltration [19,20], ion-exchange [20,21], and Donnan Dialysis [22,23]. Donnan Dialysis (DD) is a membrane process based on the cross-exchange of ions having the same electrical charge sign between two solutions through an ion-exchange membrane. Its simple operation, low installation and operating costs, and low energy consumption make Donnan dialysis (DD) a very attractive process [22,23]. Its main difference from other membrane technologies is the absence of electrical potential or pressure across the membrane, which does not require any extra energy. Calcium and magnesium ions are transported via a chemical potential gradient between the feed compartment and the receiver compartment (NaCl solution) on either side of a cation exchange membrane (CEM). Continuous ion separation is possible; however, the DD process is not applied in the industry mainly because of its slow kinetics. As known, the DD process is used for purification, concentration, and removal of some ions, such as boron, chromium, nitrates, nitrites, calcium, and magnesium, from wastewater and industrial effluents [24,25,26,27,28,29,30,31,32,33]. 

This paper presents the application of experimental designs, such as Full Factorial Design and Response Surface Methodology (RSM), using Doehlert. A Doehlert model is a powerful tool that allows data to be collected economically and efficiently as well as model its behavior to obtain mathematical functions that describe the experimental region studied, allowing statistical analysis. The Full Factorial Design is a simple systematic design style that allows for the estimation of main effects and interactions, and the main idea of RSM is to use a sequence of designed experiments to obtain an optimal response. Doehlert design was chosen in this study due to its many advantages; it allows the reduction of the number of experiments and offers high levels of each variable, which allows for obtaining maximum information about the process [34]. The Doehlert design can be considered more efficient than the central composite design or the Box-Behnken design since the efficiency of any experimental design is defined as the number of model coefficients divided by the number of experiments [35,36]. In this study, first, calcium and magnesium removal was performed with four parameters: counter-ion concentration in the receiver compartment, calcium and magnesium concentrations in the feed compartment, and the membrane nature. Then, the parameters’ effects and their interactions were studied according to a Full Factorial Design. After the determination of the most important parameters, the Response Surface Methodology using the Doehlert design was conducted to determine the optimal conditions of calcium and magnesium removal. Finally, three types of surface water destined for drinking and irrigation from different locations in Tunisia were tested to validate the suitability of Donnan Dialysis for calcium and magnesium removal.

## 2. Experiment

### 2.1. Reagents and Methods

We prepared one-component solutions from MgCl_2_·6H_2_O or CaCl_2_·2H_2_O salts (Sigma-Aldrich, St. Louis, MO, USA). These solutions constitute the feed compartment, and their final concentrations of Ca^2+^ or Mg^2+^ vary from 10 mg·L^−1^ to 200 mg·L^−1^. As a receiver, we used NaCl solution (Sigma-Aldrich) with Na^+^ concentrations varying from 0.1 mol·L^−1^ to 2.0 mol·L^−1^. All the reagents are commercially available as analytical grades and used without further purification.

The concentrations of sodium, calcium, and magnesium ions in the solution were determined by ion chromatography (Metrohm 761 compact IC ion Chromatography System; Paris—France) with a conductivity detector. The separation column used was Metrosep C4-150 (Paris—France). The eluent was composed of 2.0 mmol L^−1^ nitric acid and 0.75 mmol L^−1^ dipicolinic acid solution, with a flow rate of 0.9 mL.min^−1^, and an injection volume of 20 µL. MagIC Net is a control and database software for ion chromatography instruments that we used to perform all the calculations.

Ca^2+^ and Mg^2+^ can also be determined by other methods, such as EDTA titration, atomic absorption spectrometry (AAS), and inductively coupled plasma optical emission spectrometry (IPC-MS). As known, the method of EDTA titration is used to determine the total hardness and therefore does not allow differentiating Ca^2+^ cations from Mg^2+^. Ion chromatography, AAS, and IPC-MS are the best techniques to determine the concentrations of Ca^2+^ and Mg^2+^. In our case, ion chromatography is the most suitable method because the concentrations are often quite high, far exceeding the maximum concentrations of the two other techniques, and therefore avoid dilution operations, sources of measurement errors, and time loss. As with all our work in this manuscript, we have repeated the measurements at least three times. To evaluate the deviations on all the measurements (Tables and curves) we took the extended uncertainty values which are double the calculating standard uncertainty U for N values. Thus, each value is presented in the form of $m\pm 2\times \frac{\sigma }{\sqrt{N}}$ (*m* is the mean value and σ is the standard deviation).

### 2.2. Membranes 

Three current cationic exchange membranes CEM have been used for the Donnan Dialysis process, which are Selemion^®^ CMV, Neosepta^®^ CMS, and Neosepta^®^ CMX (supplied by Astom—Japan). Their properties are obtained according to the French standard NF X 45–200 [37] and they are listed in Table 1. Some other properties are found in the literature, especially the inter-gel fraction (f_2_) measured from the application of the microheterogeneous model [38]. This structural model is the most applied one to characterize the microstructures of homogeneous and heterogeneous ion-exchange membranes. 

Let us note already the rather particular properties of the CMS membrane, which has a good exchange capacity, the highest inter-gel fraction, but a low Ca^2+^ transport number. This is certainly due to the surface treatment that was applied to it (two thin positively charged electrolyte layers).

The chemical microstructure of these three membranes is the same and can be represented by Figure 1 [45,46]. Here, we have given only the structure of the active polymer. A PVC weave is used to support this polymer and to reinforce the mechanical properties of the membrane [47,48]. Figure 2 illustrates the different synthesis steps of these membranes [49]. Here, the matrix of synthetic ion-exchange membrane is based on (a) styrene that forms (b) polystyrene chains. The resin matrix (c) consists of polystyrene chains cross-linked with divinylbenzene. To this matrix, functional groups are added. Panel (d) shows the membrane structure of a strong cation exchange membrane containing sulfonic acid functional groups.

### 2.3. Donnan Dialysis (DD) 

Donnan Dialysis (DD) consists of cross-exchange of ions having the same electrical charge between two solutions through an ion-exchange membrane [50]. The main specificity of DD, compared to other membrane technologies, is that DD does not employ an electrical potential or pressure gradient across the membrane.

In our case, DD application in water softening makes use of only a cation-exchange membrane. The transport of Ca^2+^ and Mg^2+^ is ensured by a chemical potential gradient between the feed (natural water) and the receiver (NaCl solution) separated by the cation exchange membrane [50]. The choice of NaCl in the receptor was made mainly because it is available, cheap, fairly mobile, and does not affect the pH of the treated solutions. Two Na^+^ ions replace one Ca^2+^ ion, and electroneutrality is thus maintained. Hence, Na^+^ is commonly referred to as a compensating ion. Figure 3 presents the Donnan Dialysis scheme for the transport of Ca^2+^ and Mg^2+^ through the CEM. During the dialysis operations, the monitoring of temporal variations of Na^+^, Ca^2+^ and Mg^2+^ concentrations as well as the physicochemical characterization of the membrane allowed us to better explain the phenomenon of cation transport from one compartment to another. The samples were analyzed by ion chromatography, and the removal rate of calcium (Y_1_%), magnesium (Y_2_%), and hardness (Y_3_%) were calculated by Equations (1)–(4) as follows:(1)$$Y_{1}\left(\%\right)=\frac{{\left[{\mathrm{Ca}}^{2+}\right]}_{0}\;–\;{\left[{\mathrm{Ca}}^{2+}\right]}_{e}}{{\left[{\mathrm{Ca}}^{2+}\right]}_{0}}\times 100$$
(2)$$Y_{2}\left(\%\right)=\frac{{\left[{\mathrm{Mg}}^{2+}\right]}_{0}\;–\;{\left[{\mathrm{Mg}}^{2+}\right]}_{e}}{{\left[{\mathrm{Mg}}^{2+}\right]}_{0}}\times 100$$
where the subscripts 0 and e refer to the initial and equilibrium states, respectively.
(3)$$\mathrm{Hardness}=H=2.497\;\times \;\left[{\mathrm{Ca}}^{2+}\right]+4.118\;\times \;\left[{\mathrm{Mg}}^{2+}\right]$$
(4)$$Y_{3}\left(\%\right)=\frac{H_{0}{-H}_{f}}{H_{0}}\times 100$$
where the subscript f refers to the final hardness. 

The $\frac{\%R}{\left[M^{2+}\right]}$ ratios are the specific removal rate (SRR) (%/mg·L^−1^) expressed as the efficiency of calcium or magnesium removal ratio to the concentration of calcium or magnesium. 

### 2.4. Optimization Software

The software used in this study is NemrodW^®^, which is essential support for the practical implementation of the Experimental Research Methodology (experimental designs). A full factorial design was conducted first to determine the most important and influential parameter, then the response surface methodology design according to the Doehlert matrix was performed to obtain the optimum condition for the removal of hardness. 

## 3. Results and Discussion

### 3.1. The Preliminary Study

The delineation of the levels of each factor must be carefully considered so that the domain is neither too small nor too large, as the mathematical models may no longer fit. For this reason, a preliminary study is recommended to guide the selection of high and low levels of each factor. A preliminary study was conducted as a parameters’ pre-optimization step to define the limits of each parameter, and to better target the optimum.

#### 3.1.1. The Compensating Ion Effects

The effect of the compensating ion, here Na^+^, was examined for the removal of hardness. During the DD operation, the Na^+^ concentration varied from 0.1 mol·L^−1^ to 2.0 mol·L^−1^ when the initial concentrations in the feed compartment were equal to 100 mg·L^−1^ for each Ca^2+^ and Mg^2+^ cations, and the stirring rate was maintained at 500 rpm. The three membranes CMV, CMX, and CMS have been tested for a period of 3 h each. Figure 4a,b present the variations of Ca^2+^ (a) and Mg^2+^ (b) removal versus the compensating ion concentration. 

These two figures show that the three membranes have the same shapes of curves for both Ca^2+^ and Mg^2+^ removal rates vs. Na^+^ concentration characterized by a weak variation for low Na^+^ concentrations until 0.3 mol·L^−1^ is reached, followed by a rapid variation and finally a stabilization, or even a very slight decrease for Na^+^ concentrations exceeding 1.0 mol·L^−1^. Similar results were also reported by Vanoppen et al. [22] when studying water pre-treatment by DD for reducing multivalent cations concentrations (Ca^2+^, Ba^2+^, Fe^2+^/Fe^3+^, and Mg^2+^) before the RO process, and by Wisniewski et al. [33] who used DD to remove Ca^2+^ and Mg^2+^ with a 300 mM Na^+^ solution in the receiver compartment. 

#### 3.1.2. Effect of Calcium and Magnesium Concentrations in the Feed Compartment

The calcium and magnesium contents in natural waters were at different levels depending on the geographic location. For this reason, the effect of calcium and magnesium concentrations was explored separately in the feed compartment. We have tested a concentration variation from 10 mg·L^−1^ to 200 mg·L^−1^ of calcium and magnesium, with a Na^+^ concentration of 0.5 mol·L^−1^ and a stirring rate of 500 rpm. In addition, the same three membranes have been tested during the DD operations for the same period of 3 h. The variations of Ca^2+^ and Mg^2+^ removal rates (vertical bars) in the receiver are presented in Figure 5a,b. The curves in these figures represent the variations of the specific removal rate (SRR) $\frac{\%R}{{[M}^{2+}]}$.

Figure 5a shows that calcium flux through any CEM increases with the increase of calcium concentration from 10 mg·L^−1^ to 200 mol·L^−1^. The transport of calcium is very low at 10 mg·L^−1^ with a removal efficiency of 14.0 ± 0.9% for CMV, 12.0 ± 1.2% for CMX, and 9.0 ± 1.8% for CMS. The [Ca^2+^] concentration then increases from 10 mg·L^−1^ to 200 mg·L^−1^ and contributes to maintaining the concentration gradient of calcium high involving an improvement of the cross-ion transfer between Na^+^ and Ca^2+^. To explain the calcium diffusion through the membrane, the specific removal rate (SRR) $\frac{\%R}{{[\mathrm{Ca}}^{2+}]}$ ratio for the three membranes is presented as the curves in the same Figure 5a. According to the SRR, it seems that Ca^2+^ diffusion through the membrane was performed with the cross-exchange of Na^+^ at low concentrations, but at high concentrations, the membrane loses its performance, and, as a result, leakage without exchange is induced. Thus, it can be concluded that as the calcium concentration increases, the removal of calcium improves but in a less efficient way [29].

Figure 5b shows that magnesium flux through any CEM increases with the increase of magnesium concentration from 10 mg·L^−1^ to 200 mol·L^−1^. The transport of magnesium is very low at 10 mg·L^−1^: 13.0 ± 0.3% for CMV, 10.0 ± 0.3% for CMX, and 6.5 ± 0.5% for CMS. The [Mg^2+^] concentration then increases from 10 mg·L^−1^ to 200 mg·L^−1^ and contributes to maintaining the concentration gradient of magnesium high involving an improvement of the cross-ion transfer between Na^+^ and Mg^2+^. To explain the magnesium diffusion through the membrane, the specific removal rate (SRR) $\frac{\%R}{{[\mathrm{Mg}}^{2+}]}$ ratio for the three membranes is presented as the curves in the same Figure 5b. These curves indicate that Mg^2+^ diffusion through the membrane is performed with the cross-exchange of Na^+^ at low concentrations, but at high concentrations, the membrane loses its performance and a leakage without exchange is obtained. Thus, it can be concluded that as the magnesium concentration increases, the removal of magnesium improves but in a less efficient way [29].

#### 3.1.3. Membrane Selection

With regards to the selection of the best CEM among the three tested membranes (CMV, CMX, and CMS), we have carried out DD operations with a Na^+^ concentration of 0.5 mol·L^−1^ and a Ca^2+^ and Mg^2+^ mixture with an initial concentration of 100 mg·L^−1^ for each one. Figure 6 shows the obtained results after 3 h of operation.

According to Figure 6, the CMV membrane has the best removal rates of calcium (75 ± 0.32%) and magnesium (68 ± 0.38) simultaneously, compared to the CMX membrane, which has removal rates of almost 10–13% under those of the CMV (65 ± 0.37% and 55 ± 0.47% for Ca^2+^ and Mg^2+^, respectively), and the CMS membrane, which has almost 30% less in the removal rates of the same two cations. In fact, the CMV membrane presents a higher permeability to monovalent than bivalent cations with a high ion-exchange capacity of 2.5 meq.g^−1^, and the highest water content of 39%. These properties facilitate and accelerate the transport of calcium and magnesium from the feed compartment to the receiver compartment. CMX has a high thickness of 170 μm and a low water content of 25%, therefore, it diplays the lowest rate of calcium and magnesium removal. The CMX membrane shows a lower permselectivity of Na^+^ relative to Ca^2+^ and Mg^2+^ compared to the CMS membrane. This is certainly because of the surface treatment (see Table 1), which makes it very selective to monovalent cations and much less to divalent cations [51]. Wisniewski et al. [33] confirmed that the CMV membrane is effective compared to the CMX one for the elimination of selected anions and cations from water by means of Donnan Dialysis. Therefore, the CMV membrane has been selected for the next study.

### 3.2. Full Factorial Design

A full factorial design 2^k^ was performed to evaluate the influence of k operating parameters and their interactions with a reduced number of experiments [52]. According to the preliminary study (Section 3.1), four factors were chosen (the concentrations of the compensating ion, calcium and magnesium ions, and the cation-exchange membrane type); the retained experimental range is presented in Table 2.

The experimental response (hardness removal efficiency Y_3_) associated to a factorial design is represented by a linear polynomial model with interaction (Equation (5)): Y_H_ = b_0_ + b_1_X_1_ + b_2_X_2_ + b_3_X_3_ + b_4_X_4_ + b_12_X_1_X_2_ + b_13_X_1_X_3_ + b_14_X_1_X_4_ + b_23_X_2_X_3_ + b_24_X_2_X_4_ + b_34_X_3_X_4_(5)
where Y_H_ is the experimental response, X_i_ is the coded variable, b_i_ is the estimation of the principal effect of factor i for response Y, b_ij_ is the estimation of the interaction effect between factor i, and j for response Y.

The full factorial matrix consisting of 16 different experiments is presented in Table 3.

According to the experimental response (Table 3), the coefficients of the polynomial model were calculated allowing to predict the theoretical response through Equation (6): Y_exp_(%) = 52.81 + 6.44 X_1_ + 15.19 X_2_ + 13.81 X_3_ + 2.06 X_4_ + 1.81 X_1_X_2_ + 0.44 X_1_X_3_ − 2.56 X_2_X_3_ − 1.3 X_1_X_4_ − 0.69 X_2_X_4_ − 0.56 X_3_X_4_(6)

This model presents a coefficient of correlation higher than 0.8 (R^2^ = 0.988) implying that there is a good agreement between the theoretical response of the model and the experimental response [29,34]. The effect of factors and their interactions are shown in Figure 7a. To highlight the importance of each factor and its influence on the response compared to other factors, the Pareto analysis was carried out [53]. Results are shown in Figure 7b. 

It can be seen from Figure 7 that three factors are significant and have a positive effect on the response implying that the hardness removal efficiency improves as these factors increase. Moreover, only the [Ca^2+^]-[Mg^2+^] interaction is significant (b_23_ = −2.56) with a negative effect on the response, while all other interactions are not significant with a negligible effect on the response. 

The Pareto analysis shows that Ca^2+^ and Mg^2+^ are the most influential factors in the response with an estimated effect of 33.9% and 30.8%, respectively. Thus, these two factors contribute to about 64.7% of the response. Furthermore, these two factors are followed by the concentration of counter-ion [Na^+^] (14.6%). However, the other interactions have a negligible effect; they represent only 10.6% of the studied response.

### 3.3. Response Surface Methodology According to Doehlert

To optimize the operating conditions, the RSM was adopted as it is more economical and efficient than the traditional “one-at-a-time” method [34,36,53,54]. In this study, Doehlert design, which consists of N experiments with N = k^2^ + k + 1, was adopted (k is the parameters number; here k = 3 so N = 13. We have added two other experiments to validate the central point). It is an effective model allowing to determine the optimal conditions considering the interactions between the experimental parameters and to predict the value of the response at any point in the experimental domain by a reduced number of experiments [55,56]. The three studied factors are the initial concentrations of Mg^2+^, Ca^2+^, and Na^+^. Their experimental range and levels are presented in Table 4. Note that the Full Factorial Design allowed as to identify that the most influential parameter is X2. Thus, it is considered as the factor n° 2 in the Doehlert matrix (Table 4), allowing seven levels of variation and, therefore, a quick convergence to the optimal conditions. It is worth mentioning that the use of other models (Central Composite Design, Box-Behnken Design...) requires more than 32 experiments.

Response Y (hardness removal efficiency) is described by a polynomial model (Equation (7)):(7)$${Y=b}_{0}{+b}_{1}X_{1}{+b}_{2}X_{2}{+b}_{3}X_{3}{+b}_{11}X_{1}^{2}{+b}_{22}X_{2}^{2}{+b}_{33}X_{3}^{2}{+b}_{12}X_{1}X_{2}{+b}_{13}X_{1}X_{3}{+b}_{23}X_{2}X_{3}$$
where b_i_ is the estimation of the principal effect of factor i; b_ii_ is the estimation of the second-order effects; b_ij_ is the estimation of the interactions between factor i and factor j, and X_i_ is the coded variable.

The Doehlert matrix contains 15 experiments including three replicates at the center field [57]. The obtained results are presented in Table 5.

Using the experimental result, Y(%)_Exp_, the polynomial equation was determined (Equation (8)) allowing the prediction of the theoretical response, Y(%)_Cal_, for each experiment.
(8)$$\mathrm{YH}=74.23+3{.43X}_{1}+37{.08X}_{2}+1{.69X}_{3}+0{.17\;X}_{1}^{2}-30{.60X}_{2}^{2}-2{.36X}_{3}^{2}+2{.14X}_{1}X_{2}-4{.61X}_{1}X_{3}-2{.85X}_{2}X_{3}$$

This model presents a regression coefficient R^2^ = 0.997 greater than 0.7 and the percentage absolute errors of deviation AED (%) = 2.60%, which is less than 10%. According to these two criteria, the model can be considered valid. In addition, the variance analysis (ANOVA) presented in Table 6 shows a *p*-value less than 0.05 and an F-value higher than the critical Fischer value F_(9,5,0.05)_ = 4.77. Thus, the model is statistically significant and is therefore suitable to describe hardness removal.

According to Equation (6), the most influential factor is calcium concentration [Ca^2+^] with a positive effect on the response (+37.08), followed by magnesium concentration [Mg^2+^] (+3.46) and sodium concentration [Na^+^] (+1.69), which has a positive but moderate effect on hardness removal. This phenomenon is related to the difference in the size of ions; the hydrated ion radius of Ca^2+^ (0.412 nm) is smaller than the radius of the Mg^2+^ ion (0.429 nm) [58]. Na^+^ ions have a much smaller size in comparison to those of Ca^2+^ and Mg^2+^ ions (the radius of the hydrated Na^+^ ion is 0.358 nm [58]). This explains the increase in the mobility of Ca^2+^ and Mg^2+^ ions and causes an increase in the concentration of ions in the boundary layer of a cation-exchange membrane, which improves the removal of Ca^2+^ and Mg^2+^. It may be also due to the affinity of the CMV membrane to Ca^2+^ > Mg^2+^ > Na^+^. A similar observation indicated that the affinity order for the CMX membrane was: Ca^2+^ > Mg^2+^ > Na^+^ [41]. The CMS is a monovalent selective cation-exchange membrane (see Table 1) with a thin positively charged layer [59,60,61] applied to their both sides. These two thin layers repel polyvalent ions much more than monovalent ions. Therefore, even if the base polymer of the CMS membrane attracts more bivalent ions (electrostatic interactions between functional sites and polyvalent cations), the latter only reach the base polymer with difficulty. Thus, few polyvalent cations can cross the CMS membrane. This is confirmed by the low value of the Ca^2+^ transport number and by the large value of the inter-gel phase fraction in the presence of Na^+^ (see Table 1). On the other hand, the CMV and CMX membranes contain a low amount of a cross-linked agent and a large concentration of inorganic groups. The CMX membrane has the lowest ion-exchange capacity, and the lowest water amount, but the transport number of ions to Ca^2+^ with Mg^2+^ was better compared to CMS. The membrane CMV has the relatively high ion-exchange capacity, the highest water amount, and the transport number of ion was upper to 0.92. These parameters significantly accelerate transport from the feed to the receiver compartment [41]. 

The contour plots, shown in Figure 8 representing iso-response curves at the chosen experimental field (delimited by a circle), are used to explain the effect of factors’ interactions on hardness removal by Donnan Dialysis. 

Figure 8a presents the effect of interactions between [Ca^2+^] and [Na^+^] at a fixed [Mg^2+^]. It shows that the maximum of hardness removal is reached for [Na^+^] varying from 0.5 to 1 with [Ca^2+^] up to 55 mg·L^−1^. This is explained by the fact that the concentration gradient of the counter-ions increases, hence the cross-ion transfers between two Na^+^ ions and one Ca^2+^ to maintain electroneutrality. As shown in Figure 8b, at a fixed [Ca^2+^], hardness removal efficiency increases with an increase of [Mg^2+^] for [Na^+^] of around 0.75 mol·L^−1^. The shape of the contour plots shows little improvement in hardness removal. This was expected because the [Na^+^]-[Mg^2+^] interaction was the lowest. Figure 8c shows that at a fixed [Na^+^] hardness, removal increases as [Ca^2+^] and [Mg^2+^] increase. It is worth noting that the increase of calcium concentration contributes to maintaining the gradient concentration of calcium and magnesium high involving an improvement of cross-ion transfer.

The optimal conditions determined according to the desirability function of the NEMRODW software are 90 mg·L^−1^ for the concentration of magnesium, 80 mg·L^−1^ for the concentration of calcium, and 0.68 mol·L^−1^ for the concentration of sodium, leading to a hardness removal efficiency of 90.6 ± 5.36%.

### 3.4. Application of Different Water Types

This study aims to validate the suitability of Donnan Dialysis for Ca^2+^ and Mg^2+^ removal from natural waters. In Tunisia, we sampled three natural sources of water: lake water from Ghird el Golla, dam water from Gaafour (Oued Ettoub), and tap water from Raoued. Table 7 shows different characteristics of the water samples. The variation of water parameters can be related to the prevailing climate conditions. Indeed, salinity and temperature were lower for rainy cold periods of the year. The conductivity of tap water was the most important compared to the lake water and dam water. The increase of water conductivity is synonymous to an increase in water salinity and Total Dissolved Salts (TDS). The maximum tolerated Total Dissolved Salts (TDS) of tap water in many south Mediterranean countries like Tunisia was fixed at 1500 mg·L^−1^. TDS in water change with seasons and regions from 400 to 1400 mg·L^−1^. As these waters were collected in April 2022, this explains the amount of TDS in the three waters. The salinity and the temperature of three waters are located, respectively, between 0.7 and 0.8 g/L and between 15.9 and 18.2 °C. Ca^2+^ is most abundant in dam water, while Mg^2+^ is most abundant in lake water, owing to limestone, chalk, and gypsum deposits in the soil. As previously described, this study was performed under optimal conditions, namely 0.68 mol·L^−1^ Na^+^ concentration in the receiver compartment and CMV membrane. Despite some studies of softening natural water by DD [23,33] being limited to using NaCl (0.2 M), and therefore, may require lengthy manipulations for 5 h 30 min or may require very large amounts (2 M), the advantage of the optimization was to get the optimal concentration of compensating ion. 

We have added, in the last two lines of Table 7, the removal rates of Ca^2+^ and Mg^2+^ calculated from Figure 9. From this Figure and Table 7, it appears that dam water has the best removal rate (68% of Ca^2+^ and 58% of Mg^2+^) compared to lake water (65% of Ca^2+^ and 55% of Mg^2+^) and tap water (63% of Ca^2+^ and 53% of Mg^2+)^. This difference is attributed to the effect of accompanying ions on the transfer of a target ion to a combination of the competition for the functional groups inside the membrane and the dialytic kinetics, which reduces calcium and magnesium removal. Figure 9 summarizes the results of the analyses of four main cations (Na^+^, K^+^, Ca^2+^, and Mg^2+^) in each of the three waters used, before and after treatment with DD. Cation K^+^ has been added to the list of analyzed cations because it is found in a non-negligible way in the treated waters. Moreover, the removal rates of calcium and magnesium were relatively low in comparison to the removal rate from synthetic water. This may be due to the presence of other competitive cations (K^+^, Fe^2+^…), which may reduce calcium and magnesium transport through the membrane, leading to a decrease in calcium and magnesium removal.

It is interesting to compare the composition of lake water, dam water, and tap water after Donnan Dialysis with the cation-exchange membrane (Figure 9). As a result of the cation exchange, the molar share of Ca^2+^ ions was reduced from 140 mg·L^−1^ to 49 mg·L^−1^ in lake water (after Donnan Dialysis with the CMV membrane), from 200 mg·L^−1^ to 64 mg·L^−1^ in dam water, and from 100 mg·L^−1^ to 37 mg·L^−1^ in tap water. The molar share of Mg^2+^ ions was reduced from 72 mg·L^−1^ to 32.4 mg·L^−1^ in lake water (after Donnan Dialysis with the CMV membrane), from 60 mg·L^−1^ to 25.5 mg·L^−1^ in dam water, and from 60 mg·L^−1^ to 28.2 mg·L^−1^ in tap water. The hardness removal rate obtained in the optimization section seems higher (90.6%) compared to the rate of natural waters (68%). This difference is due to the diversity of the cationic components of the three types of water and to the salinity, which affects the chemical potential of Donnan Dialysis. These results indicate the suitability of Donnan Dialysis as a treatment that reduces hardness in natural water. 

## 4. Conclusions

This work aims to remove calcium and magnesium from Tunisian dams, lakes, and tap waters by softening water through Donnan Dialysis (DD). First, a preliminary study was conducted using three CEM membranes (CMV, CMX, and CMS) in accordance with several parameters, including the concentration of Na^+^ in the receiver compartment and the concentration of calcium and magnesium concentrations in the feed one. The CMV membrane presents a higher permeability, which facilitates and accelerates calcium and magnesium transport from the feed to the receiver compartment, and hence has the best rate of calcium and magnesium removal. The concentrations of calcium and magnesium in the feed compartment were found to be the most important parameters according to the Full Factorial Design. The optimum conditions were identified using the Response Surface Methodology (RSM) of the Doehlert design. Under optimum conditions, [Mg^2+^] = 90 mg·L^−1^, [Ca^2+^] = 88 mg·L^−1^, and [Na^+^] = 0.68 mol·L^−1^ were determined, allowing 90.6% of calcium and magnesium removal with the CMV membrane. Three different types of natural water were tested to validate the suitability of Donnan Dialysis for the removal of calcium and magnesium from real natural waters. The result indicates that the removal of calcium and magnesium from dam water is higher than that of lake water and tap water at 68% and 58%, respectively. The complexity of the water matrix with high salinity is thought to be responsible for the discrepancy. It can be concluded that the Donnan Dialysis can be competitive to other water softening technologies.

## Figures and Tables

**Figure 1 membranes-13-00203-f001:**
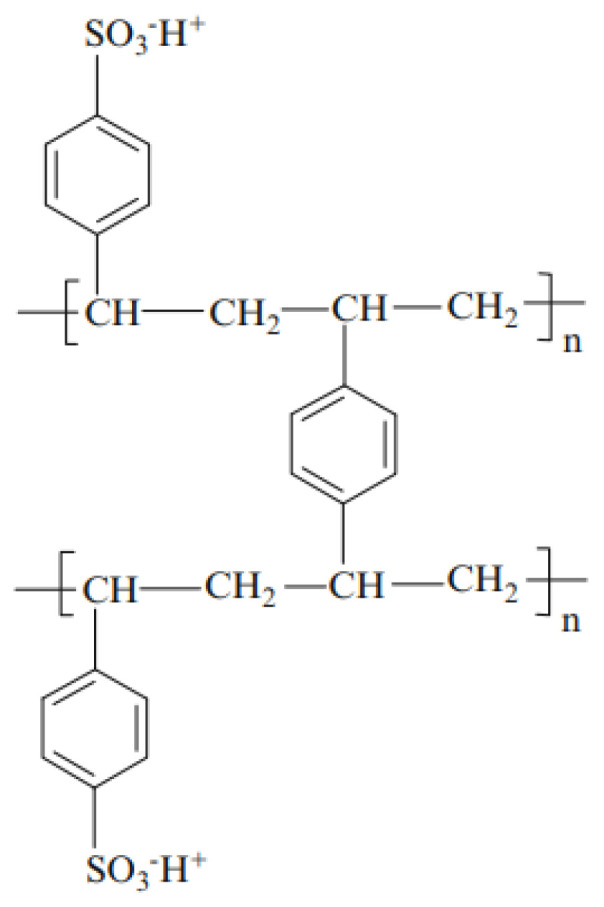
Representation of the common chemical microstructure of CMV, CMX, and CMS membranes. Here we have given only the structure of the active polymer.

**Figure 2 membranes-13-00203-f002:**
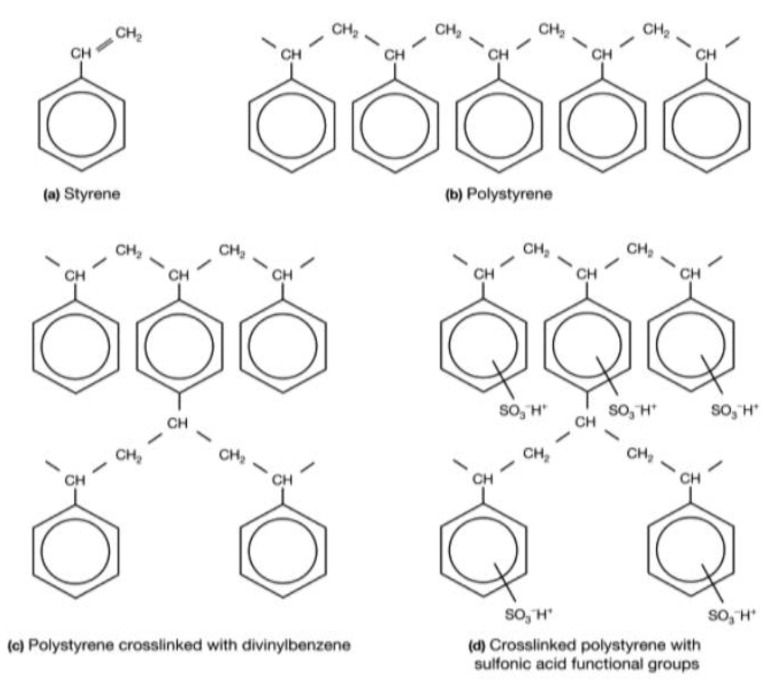
Illustration of the different synthesis steps of the active polymer of the three studied membranes.

**Figure 3 membranes-13-00203-f003:**
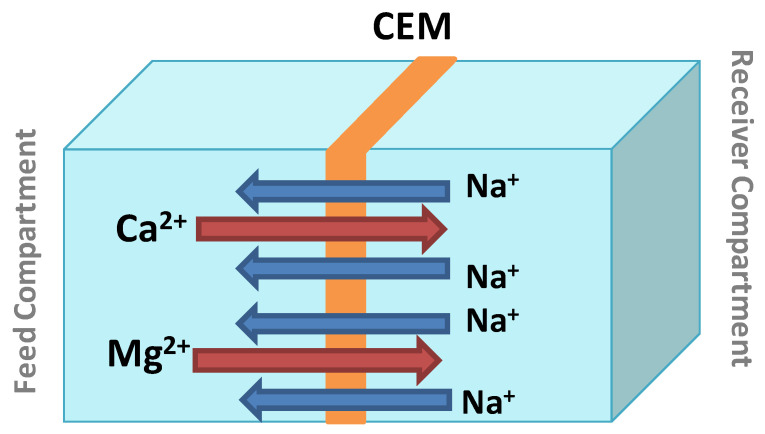
Simplified representation of calcium and magnesium removal by Donnan Dialysis.

**Figure 4 membranes-13-00203-f004:**
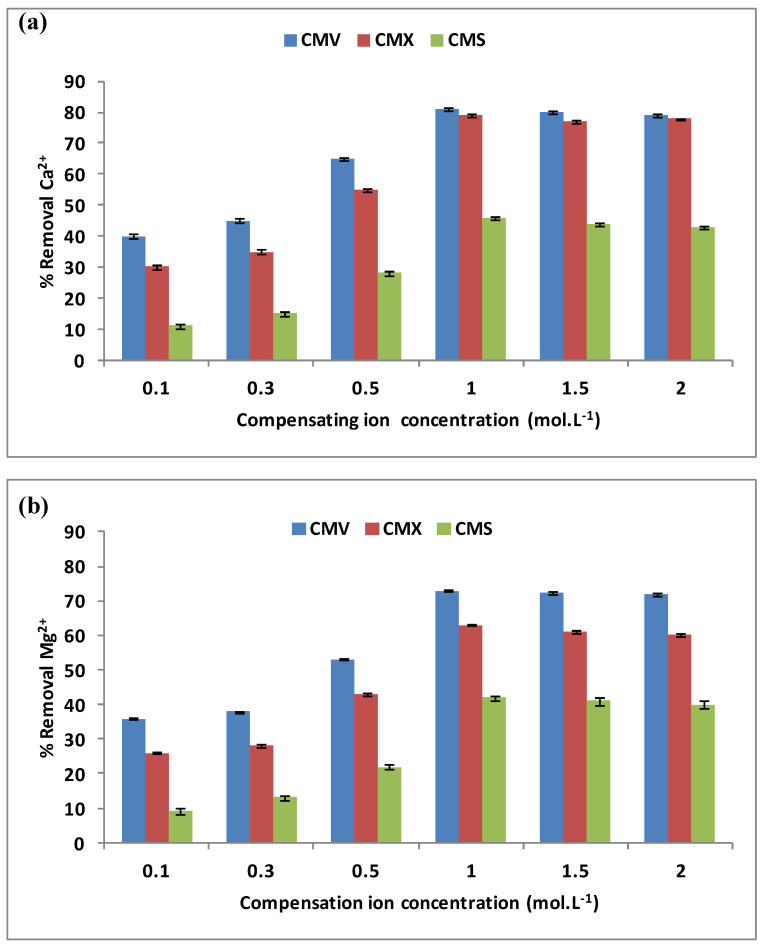
Variation of Ca^2+^ (**a**) and Mg^2+^ (**b**) removal rates with Na^+^ concentrations in the receiver. Here the initial concentrations of Ca^2+^ and Mg^2+^ in the feed compartment are equal to 100 mg·L^−1^ and the stirring rate is maintained at 500 rpm.

**Figure 5 membranes-13-00203-f005:**
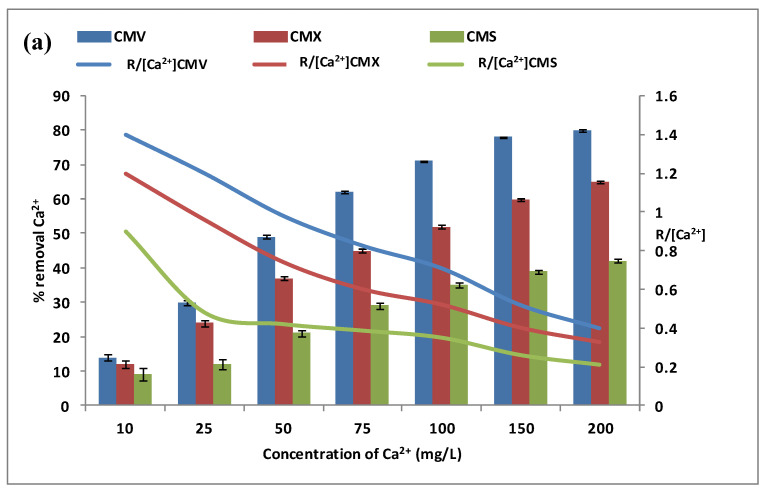
Effect of Ca^2+^ (**a**) and Mg^2+^ (**b**) concentration in the feed compartment on their removal rates (vertical bars) and on the $\frac{\%R}{{[M}^{2+}]}$ (curves). Na^+^ concentration in the receiver was equal to 0.5 mol·L^−1^, and the stirring speed was maintained at 500 rpm.

**Figure 6 membranes-13-00203-f006:**
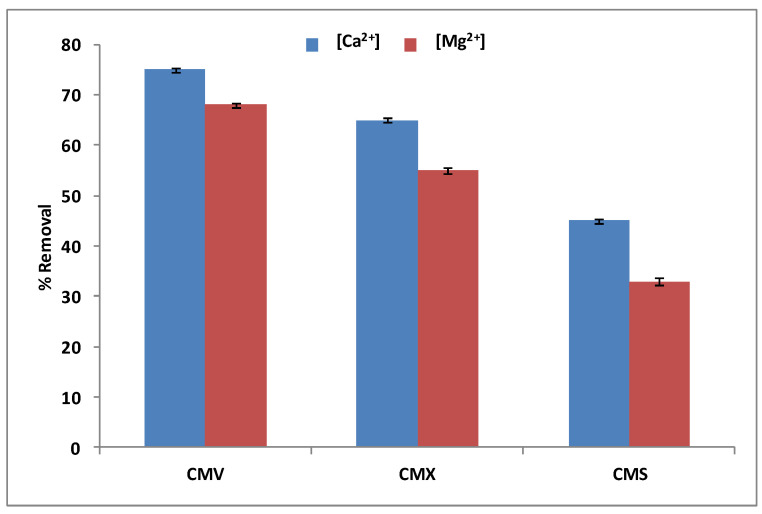
Ca^2+^ and Mg^2+^ simultaneous removal rates for the three tested cation-exchange membranes.

**Figure 7 membranes-13-00203-f007:**
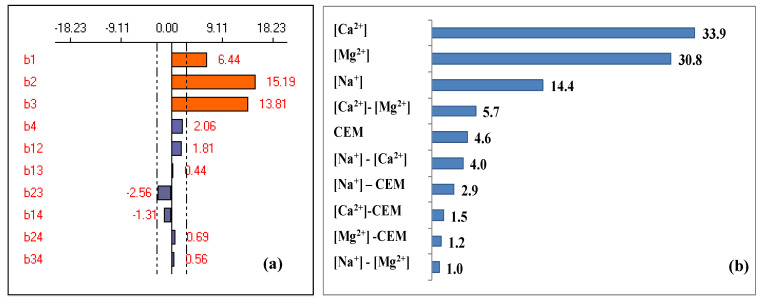
Graphical analysis of calcium and magnesium removal (**a**), and the Pareto effect (**b**).

**Figure 8 membranes-13-00203-f008:**
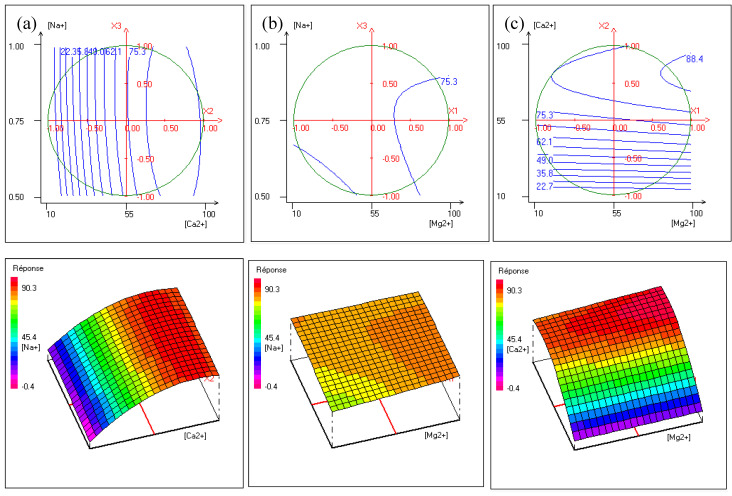
Contour plots and three dimensions plots of (**a**) calcium concentration versus magnesium concentration, (**b**) magnesium concentration versus sodium concentration, and (**c**) sodium concentration versus calcium concentration.

**Figure 9 membranes-13-00203-f009:**
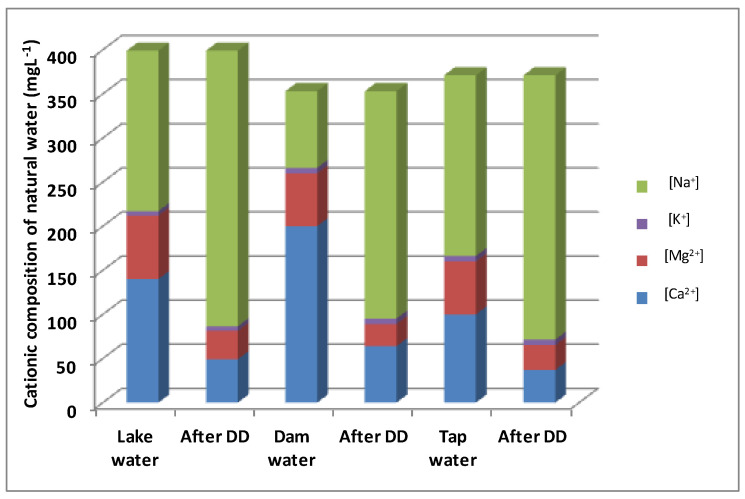
Comparison of the cationic composition of lake water, dam water, and tap water before and after Donnan Dialysis.

**Table 1 membranes-13-00203-t001:** Properties of the three cation-exchange membranes used [33,38,39,40,41,42,43,44].

Membranes	Selemion^®^ CMV		Neosepta^®^ CMX		Neosepta^®^ CMS	
**Type**	Homogeneous		Homogeneous		Homogeneous, but treated on the surface to be selective to monovalent cations.	
**Structure property**	Styrene (PS/DVB)		Styrene (PS/DVB)		Styrene (PS/DVB) & two thin positively charged electrolyte layers (unknown nature)	
**Frame backing**	PVC		PVC		PVC	
**Charged group**	Sulfonate		Sulfonate		Sulfonate	
**Exchange Capacity (meq.g^−1^) ^a^**	2.5		1.9		2.3	
**Transport number**	Na^+^, Ca^2+^ or Mg^2+^	t > 0.92	Na^+^ Ca^2+^ + Mg^2+^	t > 0.70 t > 0.28	Na^+^ Ca^2+^ + Mg^2+^	t > 0.97 t ~ 0.10
**Water Content (%) ^a^**	39		22		29	
**Inter-gel fraction (f_2_) in NaCl**	0.06		0.06–0.10		0.13	
**Permselectivity (%)**	99		98		97	
**Thickness (μm) ^a^**	130		170		150	

**^a^** Measured in this work.

**Table 2 membranes-13-00203-t002:** Experimental range and factors’ level studied in the factorial design.

Factors	Symbol	Range and Levels	
Coded Variable X_1_	[Na^+^]	−1	1
Concentration of Na^+^ (mol·L^−1^)		0.5	1.5
Coded Variable X_2_	[Ca^2+^]	−1	1
Concentration of calcium (mg·L^−1^)		10	100
Coded Variable X_3_	[Mg^2+^]	−1	1
Concentration of magnesium (mg·L^−1^)		10	100
Coded Variable X_4_	CEM	−1	1
Type of cation-exchange membranes		CMX	CMV

**Table 3 membranes-13-00203-t003:** The matrix of the Full Factorial Design.

N°	X_1_	X_2_	X_3_	X_4_	[Na^+^]	[Ca^2+^]	[Mg^2+^]	CEM	Y_3_(%)_exp_	Y_3_(%)_cal_
1	−1	−1	−1	−1	0.5	10	10	CMX	15.0	14.3
2	+1	−1	−1	−1	1.5	10	10	CMX	42.0	42.7
3	−1	+1	−1	+1	0.5	100	10	CMX	45.0	45.3
4	+1	+1	−1	+1	1.5	100	10	CMX	68.0	67.7
5	−1	−1	+1	+1	0.5	10	100	CMX	44.0	42.7
6	+1	−1	+1	+1	1.5	10	100	CMX	62.0	63.3
7	−1	+1	+1	−1	0.5	100	100	CMX	60.0	61.7
8	+1	+1	+1	−1	1.5	100	100	CMX	78.0	76.3
9	−1	−1	−1	+1	0.5	10	10	CMV	19.0	22.0
10	+1	−1	−1	+1	1.5	10	10	CMV	60.0	57.0
11	−1	+1	−1	+1	0.5	100	10	CMV	52.0	49.5
12	+1	+1	−1	+1	1.5	100	10	CMV	76.0	78.5
13	−1	−1	+1	+1	0.5	10	100	CMV	51.0	50.0
14	+1	−1	+1	+1	1.5	10	100	CMV	76.0	76.9
15	−1	+1	+1	+1	0.5	100	100	CMV	65.0	65.5
16	+1	+1	+1	+1	1.5	100	100	CMV	86.9	86.4

**Table 4 membranes-13-00203-t004:** Experimental range and factors’ levels for calcium and magnesium removal.

Factors	Range and Levels						
Coded Variable X_1_	−1		−0.5	0		0.5	1
Concentration of Mg^2+^ (mg·L^−1^)	10		33	55		78	100
Coded Variable X_2_	−0.866	−0.577	−0.287	0	0.287	0.577	0.866
Concentration of Ca^2+^ (mg·L^−1^)	16	29	42	55	68	81	94
Coded Variable X_3_	−0.816		0			0.816	
Concentration of Na^+^ (mg·L^−1^)	0.55		0.75			0.95	

**Table 5 membranes-13-00203-t005:** Doehlert Matrix and obtained results.

N°	X_1_	X_2_	X_3_	[Mg^2+^]	[Ca^2+^]	[Na^+^]	Y_3_(%)_Exp_	Y_3_(%)_Cal_
1	1.0	0.000	0.000	100	55	0.75	77.3	77.8
2	−1.0	0.000	0.000	10	55	0.75	71.5	70.9
3	0.5	0.866	0.000	78	94	0.75	88.4	86.1
4	−0.5	−0.866	0.000	33	16	0.75	16.1	18.4
5	0.5	−0.866	0.000	78	16	0.75	21.1	20.0
6	−0.5	0.866	0.000	33	94	0.75	79.7	80.8
7	0.5	0.287	0.816	78	68	0.95	79.9	81.7
8	−0.5	−0.287	−0.816	33	42	0.55	55.9	54.1
9	0.5	−0.287	−0.816	78	42	0.55	60.1	60.7
10	0.0	0.577	−0.816	55	81	0.55	82.6	84.5
11	−0.5	0.287	0.816	33	68	0.95	82.6	81.4
12	0.0	−0.577	0.816	55	29	0.95	45.5	43.7
13	0.0	0.000	0.000	55	55	0.75	74.2	74.2
14	0.0	0.000	0.000	55	55	0.75	74.2	74.2
15	0.0	0.000	0.000	55	55	0.75	74.2	74.2

**Table 6 membranes-13-00203-t006:** Analysis of variance.

Source Model	Degree of Freedom	Sum of Square	Mean of Square	F-Value	F_table_ (α = 5%)	*p*-Value
Regression	9	6831.47	759.05	158.79	4.77	0.000015
Residual	5	23.94	4.78			
Total	14	6855.42				

**Table 7 membranes-13-00203-t007:** Ionic composition of the three water types treated by DD.

	Lake Water	Dam Water	Tap Water
Turbidity (NTU)	9.40	39.59	221.14
Conductivity (µs/cm)	1603	1368	1615
pH	7.72	7.72	7.65
Temperature (°C)	18.2	17.5	15.9
Salinity	0.8	0.7	0.8
TDS (mg·L^−1^)	854	722	891
Ca^2+^ (mg·L^−1^)	140	200	100
Mg^2+^ (mg·L^−1^)	72	60	60
Na^+^ (mg·L^−1^)	181.9	86.9	204.9
K^+^ (mg·L^−1^)	5.1	6.2	6.2
Cl^−^ (mg·L^−1^)	369.2	177.5	475.7
SO_4_^2−^ (mg·L^−1^)	0.302	0.316	0.230
HCO_3_^−^ (mg·L^−1^)	488	305	396
Ca^2+^ Removal (%)	65	68	63
Mg^2+^ Removal (%)	55	58	53

## Data Availability

Not Applicable.

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
