# Peer review of "Investigation of Calcium and Magnesium Removal by Donnan Dialysis According to the Doehlert Design for Softening Different Water Types"

_membranes, 2023, doi:10.3390/membranes13020203_

Round 1
Reviewer 1 Report
In this work, the authors applied Donnan Dialysis for water softening via investigating three cation-exchange membranes (CMV, CMX, CMS). But the work is not innovative enough to be published in Membranes and its contribution to scientific research is limited. Furthermore, the introduction of the research background in this work is not sufficient. For example, the introduction part failed to introduce the research status of Donnan Dialysis.
Author Response
Dear Reviewer #1,
Please see the attached document.
Best regards
The authors

Reviewer 2 Report
Authors of the present paper aim to demonstrate the utility of DD for the water softening of several Tunisian natural waters. A praiseworthy interest and a though and experimentally well based research, anyway there are some doubts about the work itself that I´d like to be cleared by authors:
- Firstly they perform a preliminary study including all the parameters that later will be considered in the Full Factorial Design. Since the conclusions of the preliminary study appear clear and coincident with the later study, I wonder which is the utility of the Full Factorial Design later used, which finally adds more time to arrive a conclusion.
- On the other side, it appears as more important the mathematical treatment to get optimal conditions than the real separation process, so it should be somehow marked in title and aims.
- In that sense, the final study about the application of th selected membrane and experimental conditions to several water sources, seems lakc of novelty, as the capacity of DD for water softening has been extensively reported.
- Finally, which is the advantage of the preliminary study and fully completed parameter optimization for the selection of membranes and experimental parameters in a given plant? The optimization obtained is based on a previous selection of a number of membranes, what about more membranes or from other manufacturers. And also, are you sure the optimization conditions you obtained for your selected optins give real advantage for the selection (without a long previous work) fo the optimal conditions in different places or different contents sources?
Some other comments follow:
- Lines 12-13 are purely introduction and makes no sense in abstract.
- Line 36: not sure why Tunisian situation is a model for other countries. Each has different water problems.
- Line 66: a brief explanation about what means Doehlert approach is expected.
- Line 108: what means N.B. and why it is needed to remark?
- Line 109: the assert about the experimental standard in your 3 labs is not pertinent. Just indicating how many experimental repetitions have you done for each case is enough.
- Line 112: I’m favourably impressed about the way you calculate the experimental errors, as most papers just include mean and standard deviation with no more calculation (in any case, for this situation of 3 repetitions, final estimation does not change the important thing, the order of magnitude of errors).
- Line 115: CEM must be defined first time appearing, even being a well-known abbreviation
- Line 117: properties measured by authors or just manufacturer information?
- Table 1: it is not clear if four initial properties are common for all samples or only Neosepta gave information
- Table 3: don’t separate headings row from results,
- Conclusions, line 414: what to do in real applications where Ca and Mg ions concentrations cannot be changed?
- Line 424: I doubt strongly about the novelty of the conclusion on DD suitability for water softening.
Author Response
Dear Reviewer #2,
Please see the attached document.
Best regards,
The authors

Reviewer 3 Report
The manuscript shows the use of three cation exchange membranes to remove Ca2+ and Mg2+. The results are interesting. The manuscript can be improved if the fundamental structure/property relationship can be elucidated. More comments are shown below.
1. Please show the chemical structure of the three membranes, CMV, CMX, and CMS.
2. Please comment on how the membrane structures affect the removal of Ca2+ and Mg2+ ions.
Author Response
Dear Reviewer #3,
Please see the attached document.
Best regards,
The authors

Round 2
Reviewer 1 Report
It can be accepted in the current version
Reviewer 2 Report
Authors have explained properly the doubts I addressed to them and have changed the manuscript according my suggestions. So I consider the paper can be published as it.